# Maternal Supplementation with a Cocoa Extract during Lactation Deeply Modulates Dams’ Metabolism, Increases Adiponectin Circulating Levels and Improves the Inflammatory Profile in Obese Rat Offspring

**DOI:** 10.3390/nu14235134

**Published:** 2022-12-03

**Authors:** Roger Mariné-Casadó, Cristina Domenech-Coca, Anna Crescenti, Miguel Ángel Rodríguez Gómez, Josep Maria Del Bas, Lluís Arola, Noemí Boqué, Antoni Caimari

**Affiliations:** 1Eurecat, Centre Tecnològic de Catalunya, Technological Unit of Nutrition and Health, 43204 Reus, Spain; 2Eurecat, Centre Tecnològic de Catalunya, Centre for Omic Sciences (COS), Joint Unit Universitat Rovira i Virgili-EURECAT, 43204 Reus, Spain; 3Eurecat, Centre Tecnològic de Catalunya, Biotechnology Area, 43204 Reus, Spain; 4Nutrigenomics Research Group, Department of Biochemistry and Biotechnology, Universitat Rovira i Virgili, 43007 Tarragona, Spain

**Keywords:** cocoa, lactation, maternal metabolism, adiponectin, metabolic programming, offspring, MCP-1, inflammatory profile

## Abstract

High-flavonoid cocoa consumption has been associated with beneficial properties. However, there are scarce data concerning the effects of maternal cocoa intake on dams and in their progeny. Here, we evaluated in rats whether maternal supplementation with a high-flavan-3-ol cocoa extract (CCX) during lactation (200 mg.kg^−1^.day^−1^) produced beneficial effects on dams and in their normoweight (STD-CCX group) and cafeteria-fed obese (CAF-CCX group) adult male offspring. Maternal intake of CCX significantly increased the circulating levels of adiponectin and decreased the mammary gland lipid content of dams. These effects were accompanied by increased energy expenditure and circulating free fatty acids, as well as by a higher expression of lipogenic and adiponectin-related genes in their mammary glands, which could be related to a compensatory mechanism to ensure enough lipid supply to the pups. CCX consumption programmed both offspring groups towards increased plasma total adiponectin levels, and decreased liver weight and lean/fat ratio. Furthermore, CAF-CCX progeny showed an improvement of the inflammatory profile, evidenced by the significant decrease of the monocyte chemoattractant protein-1 (MCP-1) circulating levels and the mRNA levels of the gene encoding the major histocompatibility complex, class II invariant chain (Cd74), a marker of M1 macrophage phenotype, in the epididymal white adipose tissue. Although further studies are needed, these findings can pave the way for using CCX as a nutraceutical supplement during lactation.

## 1. Introduction

Adherence to unhealthy diets (i.e., diets high in calories, rich in saturated fat and high-glycemic carbohydrates, lacking in fruits and vegetables, full of fast food, and laden with high-calorie beverages and oversized portions) has increased worldwide. Although there are different causes that can contribute to the epidemic of obesity, the adoption of these unhealthy eating habits and physical inactivity are the main drivers to trigger obesity [1,2,3]. The prevalence of obesity has increased dramatically over the last four decades, with the USA and Europe being the regions with the highest prevalence (36.5% and 15.9%, respectively) [4]. Worryingly, 39 million children under the age of 5 were overweight or obese in 2020, and over 340 million children and adolescents aged 5–19 were overweight or obese in 2016 [3]. This is even more alarming because obese children can experience increased risk of hypertension and accelerated onset of type 2 diabetes and cardiovascular diseases (CVD), and childhood obesity is a strong predictor of adult obesity and is associated with disability in adulthood and premature death [3,5]. Different studies have evidenced that maternal nutritional conditions during pregnancy and lactation can produce short and long term consequences on offsprings’ health and obesity propensity [6,7,8,9] through a phenomenon called metabolic programming [10]. In this sense, it has been reported that maternal obesity or poor gestational nutrition increase the risk of obesity, CVD, and type 2 diabetes in the offspring [6,8,9]. Therefore, there is a need to underly the mechanisms that link diet with obesity, as well as to find effective nutritional strategies to tackle obesity and related diseases from the beginning of life.

Among food ingredients, cocoa, which is the primary constituent of chocolate, is one of the most consumed food products worldwide and has gained interest since it is responsible for different health benefits, mainly helping to prevent CVD and, to a lesser extent, obesity and insulin resistance [11,12]. Many studies have evidenced that its intake can improve cardiovascular health by decreasing blood pressure and LDL-cholesterol, as well as by improving vascular and platelet function through the activation of nitric oxide synthesis and the modulation of the inflammatory response. The beneficial effects of cocoa are mainly attributed to the high presence of flavan-3-ols, which is a subclass of polyphenols, including their monomeric (epicatechin and catechin) and oligomeric (proanthocyanidins) forms [11,12,13]. Related to this, in 2012 the European Food Safety Authority (EFSA) endorsed a health claim related to the effectiveness of cocoa flavan-3-ols consumption (200 mg/day) in the maintenance of normal endothelium-dependent vasodilation [14]. Despite this scientific evidence, there are scarce data concerning the effects of the consumption of cocoa-related products in pregnant and lactating women and in their offspring. In this regard, some studies have shown that chocolate rich in cocoa catechins administered to pregnant mice in a dose comparable to a daily consumption of 200 g in humans negatively affected fetal development, bone mineralization, and the function of the offsprings’ lymphoid system and kidneys [15,16,17,18]. Considering that the cocoa bioactive compounds may produce deleterious effects in the offsprings’ health, and that the diet of many pregnant and lactating women can include cocoa-related products, it is of importance to carry out animal studies before the recommendation to introduce them into the maternal diet [19].

In this regard, our group previously evaluated the metabolic programming effects of a grape seed procyanidin extract (GSPE) with a similar polyphenolic profile to that present in cocoa extracts, containing mainly epicatechin, catechin, and procyanidin dimers. We reported that, in rats, the procyanidins present in GSPE reached the placenta and fetus [20], and that this extract, administered during pregnancy and lactation to dams, exerted both beneficial [21,22] (modulation of inflammatory response, enhancement of lipid oxidation) and deleterious [21,22,23] (fat accretion, impairment of reverse cholesterol transport) metabolic programming effects in their offspring. We also showed that the supplementation with GSPE to dams exclusively during lactation increased energy expenditure and the circulating adiponectin levels of their offspring fed either a standard show (STD) or an obesogenic cafeteria diet (CAF), which resembles the MetS occurring in humans [24]; this intervention induced an adiponectin resistance-like phenotype accompanied by insulin resistance in these progeny [25].

Considering the beneficial effects described in humans for cocoa ingredients rich in flavan-3-ols and that, so far, the preclinical perinatal studies carried out with these products used very high doses when extrapolated to human consumption, we considered it of interest to evaluate the effects of the supplementation with a cocoa extract, at moderate doses, exclusively during lactation. Lactation is a less fragile and sensitive period than gestation, and this could contribute to avoiding some of the above mentioned deleterious effects tentatively associated with the intake of cocoa polyphenols during gestation. Here, we hypothesized that maternal intake of a cocoa extract during lactation, in a dose comparable to a daily consumption of about 900 mg of polyphenols in humans, would produce positive health outcomes on dams and in their offspring challenged by either balanced or unbalanced diets. Therefore, the aim of this study was to evaluate, in rats, the effects of maternal consumption of a cocoa extract at moderate doses (200 mg.kg^1^.day^−1^) during lactation on dams’ metabolism, as well as the metabolic programming effects caused by this supplementation in their offspring fed either a STD or a CAF diet from weaning to 90 days of life.

## 2. Materials and Methods

### 2.1. Cocoa Extract

Cocoa extract rich in flavan-3-ols (CocoanOX, CCX) was kindly provided by NUTRAFUR S.A. (Murcia, Spain). According to the supplier, the total phenolic content of the extract was 459 mg of catechin equivalents/g of fresh cocoa extract. The extract contained epicatechin (6.84%), catechin (0.86%), procyanidin dimer B1 (0.73%), procyanidin dimer B2 (4.15%) as well as theobromine (10.87%), maltodextrin (10%) and caffeine (1%).

### 2.2. Animals and Experimental Design

All of the procedures were approved by the Animal Ethics Committee of the University Rovira i Virgili (Tarragona, Spain), and the *Generalitat de Catalunya* (protocol code 10279). The study followed the ‘Principles of Laboratory Animal Care’, complied with the ARRIVE guidelines, and was carried out in accordance with the EU Directive 2010/63/EU for animal experiments. The animals were kept in an isolated room at 22 °C under a 12h/12h artificial light/dark cycle, and were provided with food and water ad libitum.

Eighteen 11-week-old virgin female Wistar rats (Envigo, Sant Feliu de Codines, Barcelona, Spain) were mated with adult male Wistar rats and, when pregnancy was confirmed by the presence of sperm in their vaginal smears, female rats were housed singly. After delivery (day 0 of lactation), the litter size was adjusted to ten pups per dam. From day 1, dams were randomly assigned to two groups (*n* = 9) depending on the daily treatment received: VEH group, which was orally supplemented with vehicle (low-fat condensed milk); and CCX group, which was orally supplemented with 200 mg.kg^−1^.day^−1^ of CCX dissolved in low-fat condensed milk. Doses were administered with a 1 mL syringe in a volume of 0.3–0.4 mL at the same time each day (at 09:00 a.m.), until the 21st day of lactation. The dose of CCX was equal to an intake of 1946 mg of CCX per day for a 60-kg person [26], which falls within the usual range of nutraceutical supplementation. Since the polyphenolic content of the extract was about 46%, the extrapolated dose of polyphenols administered (895 mg) is slightly below the average dietary intake of total polyphenols in Spain (1200 mg/day/person) [27]. Both groups of dams were fed with a STD (3.1 kcal/g; Teklad Global 18% Protein Rodent Diet 2018, Envigo, Sant Feliu de Codines, Barcelona, Spain). Body weight was measured every other day, and food intake was recorded every 3–4 days. On day 21 of lactation, dams were sacrificed under anesthesia (pentobarbital sodium, 60 mg.kg^−1^) after 6 h of diurnal fasting.

On postnatal day 21, male pups (at least one pup from each litter) of both groups of dams were randomly selected, weighted, single-caged and fed with either the STD or a CAF diet until the age of 90 days. This led to four experimental groups (*n* = 10–13): STD-veh, STD-CCX, CAF-veh and CAF-CCX. Both groups of rats fed the CAF diet had free access to a choice of high-energy, highly palatable foods that produce hyperphagia and mimics the MetS occurring in humans, including bacon (8–10 g), muffins (8–10 g), biscuits with cheese (10–12 g), biscuits with pâté (12–15 g) and milk with sugar (220g/L; 100 mL). The CAF diet also included carrots (6–8 g) and STD. The animals were fed ad libitum, and the food was renewed daily. Consumption of every item of the CAF diet and STD was recorded every 10 days, while body weight was monitored weekly. Animals were sacrificed at the age of 90 days and after 6 h of fasting.

In both experimental designs, blood was obtained by cardiac puncture, and plasma was separated by centrifugation and kept at −80 °C until analysis. The tissues were rapidly excised after death, weighed, frozen in liquid nitrogen and kept at −80 °C until analysis.

### 2.3. Adiposity Index

In the dams, the adiposity index was calculated by the sum of the retroperitoneal (RWAT), mesenteric (MWAT), and ovaric (OWAT) white adipose tissue depots and expressed as a percentage of the final body weight.

In the offspring, the adiposity index was calculated by the sum of the RWAT, MWAT, epididymal (EWAT), and inguinal (IWAT) white adipose tissue depots and expressed as a percentage of the final body weight.

### 2.4. Body Composition Analyses

Lean and fat mass measurements were carried out without anesthesia in both dams and in their offspring the final day of the study using an EchoMRI-700™ device (Echo Medical Systems, L.L.C., Houston, USA). The measurements were carried out in triplicate under ad libitum conditions and at 8.00 a.m. Data are expressed in relative values as a percentage of body weight.

### 2.5. Plasma Analyses

Enzymatic colorimetric kits were used to determine the circulating levels of triacylglycerols, glucose, total cholesterol (QCA, Barcelona, Spain), phospholipids (phosphatidylcholine) (Spinreact, Girona, Spain) and non-esterified free fatty acids (FFAs) (WAKO, Neuss, Germany). The plasma levels of insulin, leptin, monocyte chemoattractant protein-1 (MCP-1), and adiponectin were measured using a rat insulin ELISA kit (Mercodia, Upssala, Sweden), a rat leptin ELISA kit (Millipore, Barcelona, Spain), a rat MCP-1 ELISA kit (Thermo Scientific, Rockford, IL, USA), and a rat adiponectin ELISA kit (Millipore, Barcelona, Spain), respectively.

### 2.6. Oral Glucose Tolerance Test (OGTT)

On postnatal day 80, STD-veh, STD-CCX, CAF-veh and CAF-CCX rats were fasted overnight, and after a baseline blood collection (time 0), 2 g/kg body weight of glucose was loaded by oral gavage. Blood samples were collected from the saphenous vein at 15, 30, 60, and 120 min after the glucose challenge. Glucose at all time-points and insulin levels at baseline and at 15 and 30 min were analyzed using the commercial ELISA kits.

### 2.7. HOMA-IR and R-QUICKI Analyses

The homeostasis model assessment-estimated insulin resistance (HOMA-IR) was calculated following the formula: (Fasting Glucose level –mmol/L- × Fasting Insulin level-µU/mL-)/22.5 [28]. The revised quantitative insulin sensitivity check index (R-QUICKI) was calculated using the following formula: 1/[log insulin (µU/mL) + log glucose (mg/dL) + log FFA (mmol/L)] [29].

### 2.8. Indirect Calorimetry and Activity Measurements

Indirect calorimetry and activity analyses were performed in dams (at day 19 of lactation) and in their offspring (at day 83–84 of life) using the Oxylet Pro^TM^ System (PANLAB, Cornellà, Spain). The dams received the corresponding treatment at 09:00 a.m. Four hours after (at 01.00 p.m.), the nursing rats were separated from their pups and were transferred from their cages to an acrylic box with free access to water, but without access to food. After an initial acclimatization period of 1 h, the indirect calorimetric analyses were carried out only for 3 h (from 02:00 p.m. to 5:00 p.m.) to minimize the time of separation between them and their pups. In the offspring, these measurements were carried out with free access to water, in ad libitum conditions and for 20 h (from 12.00 p.m. to 08.00 a.m.) after allowed to become acclimated to the acrylic cages for 3.5 h (from 08:30 a.m. to 12:00 p.m.). The respiratory quotient (RQ), energy expenditure (EE), fat and carbohydrate oxidation rates, locomotor activity, and number of rearings were calculated as previously described [21].

### 2.9. RNA Isolation, cDNA Synthesis and Analysis of Gene Expression

RNA extraction, cDNA synthesis, and analysis of gene expression using real time-quantitative-PCR from mammary glands, white adipose tissue depots, liver and soleus muscle samples were performed as previously described [21], using the oligonucleotides included in Appendix A. Each PCR was performed at least in duplicate, and peptidylprolyl isomerase (*Ppia*), transferrin receptor *(Tfrc*), actin beta (*β-actin*), and hypoxanthine guanine phosphoribosyl transferase (*Hprt*) were used as reference genes.

### 2.10. Western Blot Analysis

Total and phosphorylated (p) AMP-activated protein kinase (AMPK and (p)-AMPK) in soleus muscles of the STD-veh, STD-CCX, CAF-veh, and CAF-CCX rats were determined as previously described [21]. The quantification of total T-cadherin levels in gastrocnemius muscles of these animals were determined by western blot analyses, using the same procedure described in [25].

### 2.11. Levels of Total Adiponectin in White Adipose Depots

IWAT and EWAT samples (200 mg) were homogenized as previously described [25]. The homogenate was incubated for 30 min at 4 ºC with agitation and then centrifuged at 1000× *g* for 10 min at 4 °C to remove the lipids. The levels of total adiponectin in the homogenate were measured using the rat adiponectin ELISA kit (Millipore, Barcelona, Spain) and normalized for protein content, which was determined using the Bradford protein assay [30].

### 2.12. Total Lipid Content in Mammary Glands

Lipids were extracted from mammary glands (100 mg) using the methods described in [31,32], with the modifications described in [33]. The lipid fraction obtained in this extraction were used to perform a ^1^H NMR metabolomics analysis for metabolite determination.

### 2.13. ^1^H NMR Analysis in Mammary Gland for Metabolite Determination

Lipophilic extracts of mammary glands were dissolved in 700 µL of a solution containing 0.01% tetramethylsilane (TMS) dissolved in CD_3_Cl:CD_3_OD (2:1). Samples were vortexed, homogenized for 5 min and centrifuged (15 min at 14,000× *g*). Finally, the redissolved extractions were transferred into 5 mm NMR glass tubes.

^1^H NMR measurements and analyses were performed following the procedure described by Palacios-Jordan et al. [34].

### 2.14. Statistical Analysis

Data are expressed as the mean ± SEM. Grubbs’ test was used to detect outliers. Statistical analyses were performed with SPSS Statistics 28.0 (SPSS, Inc., Chicago, IL, USA), and the level of statistical significance was set at bilateral 5%.

In dams, differences between groups (VEH and CCX) were analyzed using Student’s *t* test.

In the progeny of dams, a two-way ANOVA analysis (2 × 2 factorial design: diet (STD or CAF) x metabolic programming effect (vehicle or CCX)) was used to evaluate differences among STD-veh, STD-CCX, CAF-veh and CAF-CCX groups in energy intake, gene expression and protein data as well as in biometric, plasma and indirect calorimetry-related parameters. When the interaction between diet and metabolic programming effect was statistically significant according to two-way ANOVA, Student’s *t* test was used for single statistical comparisons between groups (i.e., the effect of metabolic programming effect within diet groups and the effect of diet within veh and CCX groups). Differences in the mRNA levels of genes involved in inflammatory response in the EWAT between CAF-veh and CAF-CCX groups were computed using Student’s *t* test.

Also in the progeny, differences in both circulating glucose and insulin levels were analyzed by repeated measures ANOVA (RM-ANOVA) with time as a within-subject factor and diet (STD or CAF) and metabolic programming effect (vehicle or CCX) as between-subject factors. When the interaction between diet and metabolic programming effect was statistically significant according to the RM-ANOVA, a subsequent RM-ANOVA was performed to analyze the effects of metabolic programming within the STD and the CAF groups.

Principal component analysis (PCA) and partial least squares discriminant analysis (PLS-DA) were performed after data normalization and autoscaling using MetaboAnalyst 3.0 software [35].

## 3. Results

### 3.1. CCX Supplementation Raised the Plasma Levels of FFAs and Total Adiponectin in Lactating Rats

Supplementation with CCX during lactation did not affect any of the biometric parameters analyzed in dams, except for a slight increase in the weight of the mammary gland (*p* = 0.096, Student’s *t*-test) (Table 1). However, the plasma levels of FFAs were significantly higher in the dams supplemented with CCX when compared to the control ones (*p* < 0.05, Student’s *t*-test) (Table 1). Except for a trend of a decrease in total cholesterol levels showed by CCX-treated dams (*p* = 0.065, Student’s *t*-test), no significant effects were detected in the other glucose and lipid metabolism-related parameters analyzed. Remarkably, CCX treatment also significantly raised the plasma levels of total adiponectin, as well as the effective production of adiponectin, meaning the total circulating levels of adiponectin corrected by the WAT weight (*p* < 0.05, Student’s *t*-test) (Table 1). In spite of that, we found no significant changes between groups for the plasma levels of high molecular weight (HMW)-adiponectin, which is the more biologically active form of this adipokine (Table 1). Of note, no differences were observed in energy intake between groups (Table 1).

### 3.2. CCX-Treated Dams Displayed Increased Energy Expenditure

In dams, the intake of CCX induced a modest, but significant, increase in energy expenditure (*p* < 0.05, Student’s *t*-test) compared to vehicle (Figure 1D), which was not due to a higher level of activity of these animals (Figure 1E). However, RQ, lipid, and fat oxidation were unchanged because of CCX supplementation during the lactation period (Figure 1A–C). The number of rearings tended to decrease in CCX-fed dams (*p* = 0.07, Figure 1F).

### 3.3. CCX Consumption during Lactation Modulated Dams’ Adiponectin Signaling and Lipid Metabolism

The lactating rats supplemented with CCX presented a significantly lower concentration of total lipids in the mammary gland (*p* < 0.05, Student’s *t*-test) (Figure 1H), accompanied by a diminished concentration of the lipid metabolites triglycerides, total omega-3 fatty acids, DHA, oleic acid, and linoleic acid (*p* < 0.05, Student’s *t*-test) (Figure 1G). All other metabolites that did not reach statistical significance are shown in Appendix A. In line with these effects, we observed a modulation of genes involved in lipid metabolism in the mammary gland, with a clear over expression of the mRNA levels of the key lipogenic genes fatty acid synthase (*Fas*), acetyl-CoA carboxylase (*Acc1)* and glycerol-3-phosphate acyltransferase (*Gpat*) (*p* < 0.05, Student’s *t* test), without changes in lipid uptake as inferred by the levels of the fatty acid transporter *Cd36* (Figure 1I). CCX dams also exhibited a significant drop in the gene encoding the hormone-sensitive lipase (Hsl) in OWAT (*p* < 0.05, Student’s *t* test), and a lower expression of *Cd36* gene in this tissue (*p* = 0.054, Student’s *t* test) (Figure 1J).

On the other hand, CCX supplementation led to a 2.3-fold increase in the expression levels of the adiponectin receptor *Adipor1* in the mammary gland (*p* < 0.05, Student’s *t* test) (Figure 1I) and a decrease of the mRNA levels of *Adipor2* in OWAT (*p* = 0.053, Student’s *t* test) (Figure 1J). Moreover, the administration of CCX during lactation also induced a significant rise of the gene encoding the endoplasmic reticulum oxidoreductin 1-like protein alpha (Ero1Lα), a key enzyme involved in HMW adiponectin secretion, in the mammary gland (*p* < 0.05, Student’s *t* test) (Figure 1I).

### 3.4. Multivariate Analysis Clearly Clustered Dams according to the Treatment Received

A total of 71 variables measured in the present study were analyzed in a PLS-DA predictive model to obtain the most relevant parameters in the differentiation of VEH and CCX groups. 2D scores plot showed a clear clustering of dams depending on the treatment that they received (Figure 2A). The quality parameters associated with the predictive model were satisfactory. When the scores of three components were represented, the degree of fit of the model to the data, which is represented by R^2^, was 0.95. Moreover, the cross-validation of this model (Q^2^) was 0.45, with a threshold of >0.4 as an acceptable value for a biological model [36]. Considering the good quality of this predictive model, we selected those variables with a coefficient mean higher than 50 to set up a PCA (Figure 2B).

Twenty-two variables were obtained and set up in a PCA, in which a 63% variance was explained when three components were represented. As shown in Figure 2C, the representation of the 22 parameters that displayed a higher relevance in the separation of both groups showed a marked clustering and, consequently, a differential response towards the consumption of CCX. All 22 normalized parameters were represented in a heat map, which showed a hierarchical clustering between both VEH and CCX groups (Figure 2D).

Among these variables, we mainly observed adiponectin metabolism-related parameters in OWAT (*Adipor2* gene expression), mammary gland (*Adipor1*, *EroL1α* and *DsbAL* gene expression) and plasma (total adiponectin levels). Moreover, lipid metabolism-related genes in OWAT (*Hsl* and *Cd36*) and mammary gland (*Acc1, Gpat, Dgat1* and *Fas*) and lipid metabolism-related parameters in plasma (FFAs) and mammary gland (omega3, PUFAs and triglycerides) were also reported.

### 3.5. CAF Diet Intake Induced a MetS-Like Phenotype in Offspring Rats

As expected [25], CAF feeding in offspring rats induced a significantly greater body weight gain and fat accretion than the STD diet, as a result of a higher energy intake (*p* < 0.05, two-way ANOVA) (Table 2). Concomitantly, these rats developed alterations associated with obesity, including dyslipidemia and disturbed glucose metabolism, as well as marked hyperleptinemia (*p* < 0.05, two-way ANOVA) (Table 2, Figure 3A,B). Likewise, the CAF diet intake led to an increase in carbohydrate oxidation and locomotor activity of the animals, which reached significance when compared to STD animals (*p* < 0.05, two-way ANOVA) (Figure 3E,G). Furthermore, plasma total adiponectin levels were significantly higher in CAF-fed offspring rats than in their STD-fed counterparts (*p* < 0.05, two-way ANOVA) (Figure 4A), while the effective production of adiponectin displayed the opposite pattern, being significantly reduced as a result of CAF feeding (*p* < 0.05, two-way ANOVA) (Figure 4C). Finally, as it has been observed in diet-induced obese rodents [37], overall, CAF-veh animals displayed decreased adiponectin/leptin ratio than the STD-fed groups (*p* < 0.05, Student’s *t* test) (Table 2).

### 3.6. The Offspring of Lactating Dams Supplemented with CCX Displayed Decreased Liver Weight and Lean/Fat Ratio

Male offspring of rats supplemented with CCX during lactation showed lower liver weight than their counterparts (*p* < 0.05, two-way ANOVA) (Table 2). Additionally, maternal CCX supplementation led to lower soleus muscle weight and a lower lean/fat ratio of the offspring (*p* < 0.05, two-way ANOVA) (Table 2), a result mainly attributed to the STD groups. Regarding plasma parameters, a significant interaction between diet and metabolic programming was reported for the postprandial glucose levels after the OGTT (*p* < 0.05, RM-ANOVA). The subsequent RM-ANOVA analysis performed to evaluate the metabolic programming effects of CCX within the diet groups, showed a trend towards an interaction between time and metabolic programming effect (*p* = 0.057, RM-ANOVA) and towards a metabolic programming effect (*p* = 0.089, RM-ANOVA) only in CAF-fed animals (Figure 3A). These results were mainly attributed to the numerically lower circulating levels of glucose exhibited by CAF-CCX rats in comparison with their CAF-veh counterparts, especially at 60 min. (Figure 3A). These findings were not accompanied by significant changes in the circulating levels of insulin over the OGTT (Figure 3B). In addition, triglyceride plasma levels tended to be lower in the offspring of CCX-fed rats than in the control groups (*p* = 0.080, two-way ANOVA) (Table 2), without any changes in the other analyzed parameters (Table 2). The results from indirect calorimetry measurements did not reflect any modification induced by the maternal intake of CCX in their offspring (Figure 3C–H).

### 3.7. CCX Intake during Lactation Induced a Clear Metabolic Programming Effect in the Adiponectin Plasma Levels of the Offspring

Supplementation with CCX during lactation led to a significant rise in plasma total adiponectin levels in both STD-fed and CAF-fed offspring (*p* < 0.05, two-way ANOVA) (Figure 4A). When we looked at the active form of adiponectin, we found numerically higher values of circulating HMW-adiponectin levels in the offspring from CCX-fed dams, although this difference was not significant (*p* = 0.084, two-way ANOVA) (Figure 4B). However, the increase in plasma adiponectin induced by CCX was not translated into a higher effective production of adiponectin (Figure 4C). Then, we analyzed the concentration of total adiponectin in different fat depots. Although we did not observe any differences in EWAT and IWAT separately, when we considered the sum of these two adipose tissues, we detected that adiponectin tissue concentration tended to decrease in the animals coming from dams exposed to CCX (*p* = 0.081, two-way ANOVA) (Figure 4D). In addition, we found a significant interaction between diet and metabolic programming for the adiponectin/leptin ratio (*p* < 0.05, two-way ANOVA) (Table 2). Thus, this ratio significantly increased in STD-CCX vs. STD-veh group (*p* < 0.05, Student’s *t* test), which is associated with metabolic health and reduced CVD risk [37], an effect that was not observed in their CAF-CCX counterparts.

No differences were obtained in skeletal muscle T-cadherin and (p)-AMPK protein levels either in STD-CCX or in CAF-CCX rats in comparison with their counterparts (Figure 4E,F). Regarding the expression levels of genes related to adiponectin signaling in key tissues, we found a significant interaction between diet and metabolic programming effect of CCX in *Adipor2* mRNA hepatic levels (*p* < 0.05, two-way ANOVA) (Figure 4G), although when the main effects were analyzed for each factor independently, we did not find any changes induced by CCX maternal supplementation. The expression of the genes encoding adiponectin receptors 1 and 2 in muscle and IWAT, as well as the expression of *Adipor1* in liver, were unchanged by CCX maternal intake (Figure 4G). We also found a significant interaction between diet and metabolic programming effect in the expression levels of muscular *Acc2*, a rate limiting enzyme in de novo lipogenesis, which rise induced by CAF feeding was completely prevented by CCX maternal supplementation (*p* < 0.05, two-way ANOVA) (Figure 4G). Finally, we found that, overall, the mRNA levels of the key regulator of lipid metabolism *Pparα* were significantly increased in the muscle of STD-CCX and CAF-CCX offspring (*p* < 0.05, two-way ANOVA) (Figure 4G).

### 3.8. Maternal CCX Supplementation Restored the Plasma Levels of the Inflammatory Marker MCP-1 in CAF-fed Offspring

Interestingly, we found a significant interaction between diet and the metabolic programming effect of CCX in the circulating levels of the pro-inflammatory chemokine MCP-1 (*p* < 0.05, two-way ANOVA) (Figure 5A). As expected [25], CAF-veh animals displayed increased circulating levels of this pro-inflammatory biomarker in comparison with their STD-veh counterparts (*p* < 0.05, Student’s *t* test) (Figure 5A). A subsequent pairwise analysis revealed that the MCP-1 serum levels were markedly diminished by the maternal CCX treatment only in CAF-fed offspring rats, restoring the levels observed in STD-fed rats (*p* < 0.05, Student’s *t* test) (Figure 5A). To achieve a deeper understanding of the mechanisms implicated in this remarkable programming effect, we analyzed the expression levels of key genes related to inflammation and MCP-1 pathway in the EWAT of the CAF-fed offspring (Figure 5B). Specifically, we analyzed the mRNA levels of genes used as markers for both the M1 classical activated phenotype (*Mcp1*, *Tnfα*, *Emr1*, *iNOS* and *Cd74*) and the alternatively activated M2 phenotype (*Mgl1*, *Il10* and *Mrc1*) of macrophages. Although the expression levels of *Mcp1* were not significantly decreased by CCX supplementation in the EWAT of the CAF-CCX animals, we observed a large and significant drop in the expression levels of the gene encoding Cd74 because of maternal CCX intake, which were half of those reported for the CAF-veh rats (*p* < 0.05, Student’s *t* test) (Figure 5B). We did not find additional changes between the CAF-veh and CAF-CCX groups in the expression of the other inflammation-related genes analyzed in this tissue (Figure 5B).

## 4. Discussion

In this work, we demonstrated that supplementation with a cocoa extract rich in flavan-3-ols during lactation considerably increased plasma adiponectin and FFA levels in dams, while it modulated the mammary gland metabolism by inducing an increase in the expression of lipogenic and adiponectin-related genes, and a reduction of lipid concentration. In fact, the multivariate analysis performed on the 71 parameters analyzed in dams, including biometric, biochemical, and molecular variables in blood, mammary gland, liver and OWAT, showed a clear clustering between VEH and CCX dams, mostly highlighting adiponectin and lipid metabolism-related parameters in the mammary gland as the most relevant variables contributing to the differentiation of both groups.

Cocoa has been shown to stimulate adiponectin production in vitro [38] and to raise adiponectin circulating concentrations in both rodents [39] and humans [40,41]. A possible mechanism implicated in this effect is the activation of Ppar-ɣ by cocoa polyphenols, which has been observed in rats [42], since Ppar-ɣ is the major positive regulator of adiponectin expression [43]. However, we did not find changes either in the expression levels of adiponectin in OWAT or in genes related to adiponectin secretion, such as Ero1-Lα and DsbA-L, which are all regulated by Ppar-ɣ. We cannot rule out that higher adiponectin expression and secretion could be found in other fat depots, since the higher effective production of adiponectin in CCX-treated dams suggests greater adiponectin secretion by white adipose tissue in these rats. In addition, inflammatory molecules act as negative regulators of adiponectin secretion, so the anti-inflammatory properties associated with cocoa consumption [13] could also contribute to explaining the reported increase in the maternal circulating adiponectin levels.

The lower content of lipids detected in the mammary gland of our lactating dams supplemented with CCX probably reflects a lower concentration of milk lipids, since this has been consistently described in the literature [44,45]. Other polyphenolic extracts, such as a green tea extract, have been shown to decrease milk fat concentration in dairy cows [46]. In fact, our group has previously published similar results induced by the intake of GSPE in lactating rats [25]. However, a great number of studies relate the intake of various sources of polyphenols to an increase in milk production and milk fat concentration [47,48,49], or to a null effect in these parameters [50,51], mainly in ruminants. Thus, it is probable that these discrepancies reflect the genetic and metabolic differences between animal species as well as differences in the polyphenol’s composition and concentration of the studied extracts.

The lactating mammary gland requires a major supply of fatty acids to meet the triglycerides synthesis demands, since approximately 98% of milk lipids are composed of triglycerides [52]. The lipid substrates used by the mammary gland are dietary fat, FFAs released by the adipose tissue and lipids synthesized de novo by the tissue itself. Thus, the raise in plasma FFAs observed in the CCX-supplemented lactating rats could be a compensatory mechanism by which these animals try to counteract the decreased lipid content of the mammary gland, providing extra fat sources to this tissue. Although we did not find differences in the expression of the fatty acid transporter CD36 in the mammary gland of CCX-supplemented rats, other membrane transporters could be involved in this process [53,54]. Alternatively, the increased circulating FFAs levels induced by CCX intake could be the result of a higher fatty acid uptake from triglyceride-rich lipoproteins and consequently a higher spillover, meaning the escape of some lipolytic products resulting from lipoprotein lipase activity, which is known to represent a significant additional source of plasma FFAs [55], especially when coming from chylomicrons.

The higher expression of lipogenic genes *(Fas*, *Acc1*, *Gpat*) observed in the mammary gland of dams supplemented with CCX could be also in line with the decreased concentration of lipids in this tissue, acting as a compensatory mechanism to ensure enough lipid supply to the offspring. In addition, adiponectin has been shown to be essential for the normal function of the mammary gland while its overexpression promotes the uptake of circulating FFAs and triglycerides by the lactating mammary gland [56]. Thus, the induction of adiponectin plasma levels observed in CCX-treated dams, accompanied by the sharp increase of its receptor *Adipor1* in the mammary gland and the lower expression of *Adipor2* in OWAT, suggest that adiponectin could be also acting as a facilitator of fatty acids transferring to the mammary gland. Interestingly, *Ero1-Lα* expression increased in the mammary gland of CCX-supplemented dams. This did not lead to higher levels of plasma HMW-adiponectin in dams, so it can be speculated that CCX stimulated the secretion of HMW-adiponectin into the milk of lactating rats. The increased EE observed in the lactating dams supplemented with CCX would agree with our hypothesis pointing towards increased milk production as a result of higher suckling activity of the offspring, in an attempt to counteract the reduced milk fat concentration.

Lactation is a critical period for development and, therefore, maternal nutrition interventions during this period can induce metabolic programming effects, affecting health outcomes of the offspring in adult life. Interestingly, the increase in adiponectin plasma levels reported in dams supplemented with CCX during lactation was also observed in their STD-CCX and CAF-CCX adult offspring, indicating a clear programming effect related to this adipocytokine. In this sense, a previous study performed by our group had already shown the effectiveness of GSPE consumption in producing such programming effect, raising the circulating levels of total and HMW-adiponectin in the offspring of dams that were treated with this polyphenol extract during lactation [25]. Therefore, it is plausible to hypothesize that the common flavan-3-ols present in both extracts (mainly epicatechin, catechin and procyanidin dimers) would be the main drivers of this metabolic programming effect. Although we could not collect the milk of the nursing rats and, therefore, we cannot establish a direct association between changes in dams’ milk composition in CCX dams and the programming effects in their offspring, we attempted to unravel some of the underlying mechanisms implicated in the adiponectin-related effects observed in the progeny of the CCX-supplemented dams. Our results do not suggest increased adiponectin production by WAT. Nevertheless, the concentration of adiponectin tended to decrease in EWAT and IWAT considered as a whole, indicating a possible stimulation in adiponectin release by WAT that could partially explain the higher plasmatic concentration of adiponectin found in the offspring from dams exposed to CCX. On the other hand, the higher circulating adiponectin levels reported in both STD-CCX and CAF-CCX offspring did not seem to be associated with a lower clearance of adiponectin from circulation, as no changes were observed in skeletal muscle T-cadherin content, a protein with a key role in the regulation of circulating levels of adiponectin [57]. Additional analyses focused on the quantification of total or other forms of adiponectin, or on other mechanisms that could be involved in the higher circulating levels of this adipocytokine, such as hepatic clearance from the bloodstream [58,59] or autophagy-mediated adiponectin degradation [43], would be useful to shed light on this issue.

A cafeteria diet contains highly palatable and energy-dense foods that predominate in Western society. This diet induces hyperphagia, obesity, insulin resistance and hypertriglyceridemia [24,60], as was reported in the present study. It has been widely described that adiponectin possesses beneficial effects on glucose and lipid metabolisms, acting as an insulin sensitizer by reducing glucose and lipid production in the liver and promoting the utilization of glucose and fatty acids by the skeletal muscle, lowering blood glucose levels, and improving the circulating lipid profile [61]. In the present study, the slight improvement in postprandial glucose levels observed in CAF-CCX offspring could be tentatively related to the increased circulating levels of adiponectin and would agree with previous studies in which cocoa proanthocyanidin extracts prevented impaired glucose tolerance in high-fat fed mice [62,63]. These results clearly differed from those obtained in a previous study performed by our group, in which GSPE maternal consumption during the lactation period provoked deleterious effects, namely an adiponectin resistance-like phenotype accompanied by insulin resistance, in their STD and CAF progeny [25]. Despite the similar profile of polyphenols between both extracts, the different concentration of these flavan3-ols between CCX and GSPE as well as the presence in CCX of the methylxanthines theobromine and caffeine, which possess several health benefits [64,65], could account for the aforementioned differential programming effects on glucose metabolism.

CCX offspring also exhibited a lower lean/fat ratio, which is considered an indicator of an unhealthy body composition [66]. This result is in line with the lower soleus muscle weight displayed by these animals, an effect that was numerically more evident in the normoweight STD group. Despite observing this apparent detrimental effect in the soleus muscle, the phosphorylated levels of AMPK, which is considered an essential mediator of adiponectin actions on lipid metabolism [67], as well as the expression levels of lipid metabolism-related genes reported in this tissue did not indicate any metabolic disturbances. Intriguingly, opposing roles of adiponectin in the regulation of muscle mass health are found in the bibliography. Despite the fact that several publications have described the role of adiponectin in promoting muscle regeneration [68,69] and proteolysis inhibition, other in vitro and clinical studies found an association between higher adiponectin levels and muscle wasting [70,71]. Further studies would be required in order to elucidate the pathophysiological significance of these findings.

A low adiponectin/leptin ratio (<1), which is characterized by a lower secretion of adiponectin in relation to leptin, has been described as a marker of dysfunctional adipose tissue and it is associated with cardiometabolic risk in humans, reflecting the obesity-related disturbances in the adipose tissue [72,73]. This ratio has been described to negatively correlate with insulin resistance and several markers of low-grade chronic inflammation in obese people, being considered a predictive biomarker of MetS [73]. As was previously described for diet-induced obese mice [37], in the present study, CAF feeding led to a strong decrease of adiponectin/leptin ratio. Intriguingly, in the present study only the offspring of rats supplemented with CCX that were fed with the STD diet developed a sharp increase in the adiponectin/leptin levels compared to the STD-veh group. Although an adiponectin/leptin ratio between 1 and 4 has been positively associated with metabolic health and reduced CVD risk [37], whether the reported increase would be translated into positive health outcomes in the normoweight offspring would require long-term studies focused on aged progeny, as it is well known that age is an independent risk factor for CVD [74].

Even though the CAF-CCX rats did not display a higher adiponectin/leptin ratio than their CAF-VEH counterparts, the analysis of the inflammatory profile showed a significant decrease in plasma MCP-1 levels and a sharp down regulation of the EWAT mRNA levels of *Cd74* in the CAF-CCX progeny. Remarkably, the lower levels of these pro-inflammatory biomarkers would indicate that the obese progeny of dams treated with CCX have a more favorable inflammatory profile compared to the CAF offspring of vehicle-treated dams, suggesting that they are more protected against the pro-inflammatory state that characterizes obesity and MetS. It has been widely described that polyphenols, included those present in cocoa, have the capacity to improve the overall inflammatory state, modulating several inflammatory parameters related to obesity and MetS, including a decrease in the circulating levels of pro-inflammatory cytokines and chemokines [11,13,75]. In addition, it has been shown that maternal intake of different polyphenols such as genistein, resveratrol, epigallocatechin-3-gallate and anthocyanins during pregnancy and/or lactation can favorably program the offspring towards a better inflammatory profile [19,75,76]. Related to this issue, our group previously demonstrated that the high fat diet-fed young offspring of dams supplemented with GSPE during pregnancy and lactation displayed decreased circulating levels of MCP-1 [22]. Considering the similar profile of polyphenols between GSPE and CCX, our findings suggest that these flavan-3-ols present in both extracts would be responsible for the beneficial programming effects regarding MCP-1 plasma concentrations. Although both preclinical and clinical studies have shown that the supplementation of cocoa products rich in polyphenols are able to reduce the circulating levels of MCP-1 [38,39,41], as far as we know, this is the first study reporting that the maternal intake of a cocoa extract exclusively during lactation produces such programming effect in their adult offspring induced to MetS.

The decrease in the MCP-1 plasma levels observed in the CAF-CCX animals could be attributed, at least in part, to the increased circulating levels of adiponectin observed in these rats, taking into account the anti-inflammatory activity exerted by adiponectin [77,78]. The inflammatory state of white adipose tissue is, in part, the result of the balance between the two subtypes of macrophages infiltrate in this tissue, namely classically activated macrophages, also called M1 macrophages, and alternatively activated macrophages, or M2 macrophages [79]. On one hand, obesity causes an increase in the number of M1 macrophages, which contribute to trigger the chronic pro-inflammatory response that contributes to the pathogenesis of obesity-induced insulin resistance [79,80]. On the other hand, M2 macrophages have a crucial role in maintaining the insulin sensitivity of adipocytes (via secretion of IL-10) and contribute to the improvement and repair of vascular function through their anti-inflammatory properties [79,81]. Adiponectin exerts its anti-inflammatory effects in part by priming monocytes into M2 macrophages, promoting the expression of anti-inflammatory M2 markers (Arg-1, Mgl-1, IL-10), and inhibiting the differentiation and classical activation of M1 macrophages by down regulating pro-inflammatory cytokines (TNF-a, MCP-1 and IL-6), thus modulating the macrophage-mediated inflammation towards a more favorable profile [82,83]. In the present study, the sharp down regulation of the expression of the gene encoding Cd74, a molecular marker of M1 macrophage phenotype involved in antigen presentation in different immune cells [84], observed in the EWAT of the CAF-CCX rats, would suggest and adiponectin-driven anti-inflammatory programming effect of CCX in this visceral white adipose tissue. Gene expression data do not always match protein levels. Thus, further research focused on protein and/or flow-cytometry-based analyses in white adipose depots or in other tissues containing also cell types that are sources of MCP-1, such as monocytes/macrophages, fibroblasts, astrocytes, and epithelial, endothelial, smooth muscle and microgial cells [85], would contribute to shed light on the mechanisms that are responsible for the aforementioned CCX programming effects.

All of these metabolic programming effects resulting from maternal CCX supplementation, driven by milk intake can be explained by different mechanisms. First, polyphenols have been shown to induce epigenetic modifications including DNA methylation, histone acetylation and miRNAs regulation [86], which could explain these persistent metabolic effects in the adult age of the offspring. Interestingly, breast milk miRNAs related to adiponectin function have been described to be altered by maternal diet in rats [87]. Second, oxidative stress plays a crucial role in the developmental programming of MetS [88] and, thus, the metabolic programming effects associated to antioxidant molecules such as polyphenols can be explained by a reduction of oxidative stress in early life. Third, polyphenols are known to modify gut microbiota [89], which has also been related to metabolic imprinting effects [88]. Finally, alterations in milk composition during lactation can produce profound metabolic programming effects. Thus, although a reduction of milk fat concentration has been related to detrimental health effects in the offspring [90], the slight decline of lipids content in the mammary gland observed in CCX-fed dams was probably not enough to induce harmful metabolic alterations in the offspring.

## 5. Conclusions

To sum up, we demonstrated that the administration of CCX in dams during lactation significantly increased the circulating levels of adiponectin and decreased their mammary gland lipid content. These effects were accompanied by increased energy expenditure and circulating free fatty acids, as well as by a higher expression of lipogenic *(Fas*, *Acc1*, *Gpat*) and adiponectin-related genes (*Adipor1* and *Ero1-Lα*) in their mammary glands, which could be related to a compensatory mechanism to ensure enough lipidic supply to the pups. The profound modulation of the dams’ metabolism was translated into programming effects in their both STD-CCX and CAF-CCX progeny, which displayed increased circulating levels of adiponectin and decreased liver weight and lean/fat ratio. In addition, CCX consumption programmed dams’ CAF-fed obese offspring towards an improvement of the adipose tissue inflammatory profile, evidenced by the significant decrease of MCP-1 circulating levels and the sharp down regulation of the gene encoding Cd74 in EWAT. Overall, the above commented metabolic programming effects resulting from maternal CCX supplementation, driven by milk intake, evidenced that despite the slight decline of lipid content in the mammary glands observed in CCX-fed dams, which would suggest lower milk lipid content and supply to pups, no evident detrimental metabolic alterations were observed in the offspring. Altogether, the present study remarks on the beneficial effects of cocoa consumption at a moderate dose during the lactating period in both dams and progeny. However, additional dose-response studies testing lower doses of this cocoa extract should be performed in order to rule out some of the potential harmful effects observed in the present study (i.e., decreased lipid content in mammary glands and lower lean/fat ratio in the offspring) to pave the way to recommend supplementation of flavan-3-ols-rich cocoa extracts in the maternal diet.

## Figures and Tables

**Figure 1 nutrients-14-05134-f001:**
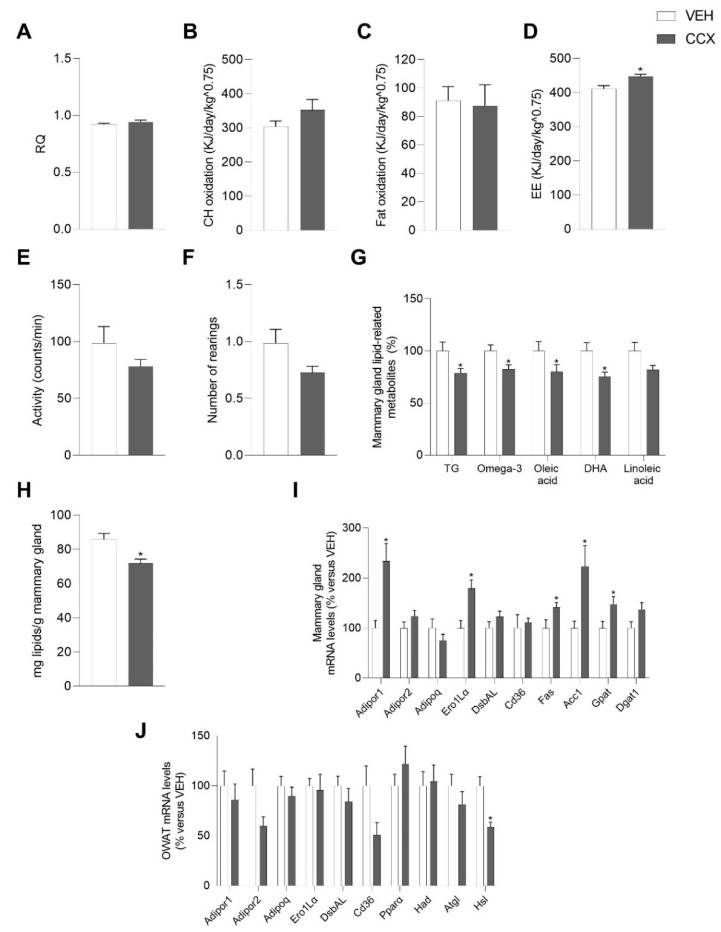
Respiratory quotient (RQ) (**A**), carbohydrate (CH) oxidation (**B**) fat oxidation (**C**) and energy expenditure (EE) (**D**), locomotor activity (**E**) and number of rearings (**F**) measured by indirect calorimetry analyses at day 19 of lactation of dams treated with either CCX (CCX group) or vehicle (VEH group) during lactation. Expression levels of adiponectin metabolism-related genes in mammary gland (**I**) and OWAT (**J**) as well as the concentration of total lipids (**H**) and lipid-related metabolites (**G**) in the mammary gland are also shown. Data are the mean ± SEM (*n* = 8–9). * Effect of CCX treatment (Student’s *t* test, *p* < 0.05). CCX, cocoanox extract; TG, triglycerides; DHA, docosahexaenoic acid; *Adipor1*, adiponectin receptor 1; *Adipor2*, adiponectin receptor 2; *Adipoq*, adiponectin; *Ero1-Lα*, endoplasmic reticulum oxidoreductin 1-like protein alpha; *DsbAL*, disulfide-bond-A oxidoreductase-like protein; *Cd36*, fatty acid translocase, homologue of CD36; *Fas*, fatty acid synthase; *Acc1*, acetyl CoA carboxylase 1; *Gpat*, glycerol-3-phosphate acyltransferase; *Dgat1*, diacylglycerol O-acyltransferase 1; OWAT, ovarian white adipose tissue; *Pparα*, peroxisome proliferator-activated receptor alpha; *Had*, hydroxyacyl-CoA dehydrogenase; *Atgl*, adipose triglyceride lipase; *Hsl*, hormone-sensitive lipase.

**Figure 2 nutrients-14-05134-f002:**
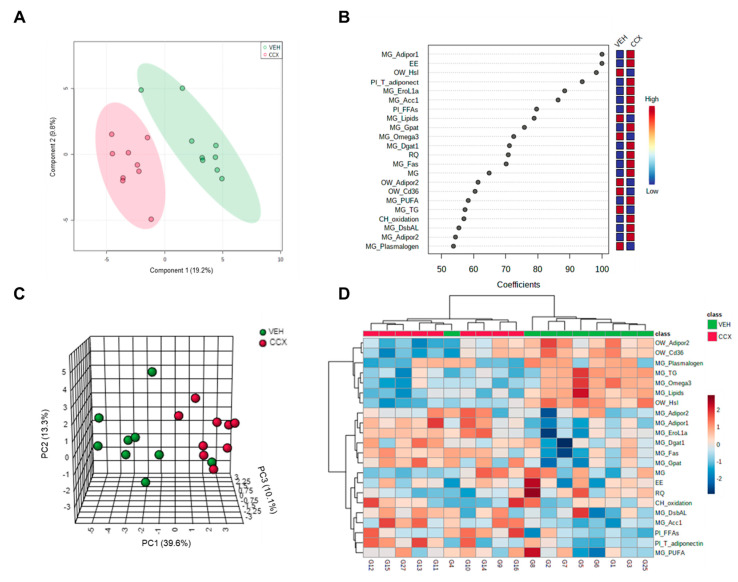
A total of 71 parameters were used to perform a PLS-DA predictive model (**A**). Those parameters with a coefficient mean higher than 50 (**B**) were analyzed in a PCA multivariate analysis (**C**). 22 normalized parameters were also represented in a Heat Map (**D**). CCX, cocoanox extract; MG, mammary gland; *Adipor1*, adiponectin receptor 1; EE, energy expenditure; OW, ovaric white adipose tissue; *Hsl*, hormone-sensitive lipase; Pl, plasma; T, total; Adiponect, adiponectin; *EroL1α*, endoplasmic reticulum oxidoreductin 1-like protein alpha; *Acc1*, acetyl CoA carboxylase 1; FFAs, free fatty acids; *Gpat*, glycerol-3-phosphate acyltransferase; *Dgat1*, diacylglycerol acyltransferase 1; RQ, respiratory quotient; *Fas*, fatty acid synthase; *AdipoR2*, adiponectin receptor 2; *Cd36*, fatty acid translocase, homologue of CD36; PUFA, polyunsaturated fatty acid; TG, triglycerides; CH, carbohydrate; *DsbAL*, disulfide-bond-A oxidoreductase-like protein; PC, principal component.

**Figure 3 nutrients-14-05134-f003:**
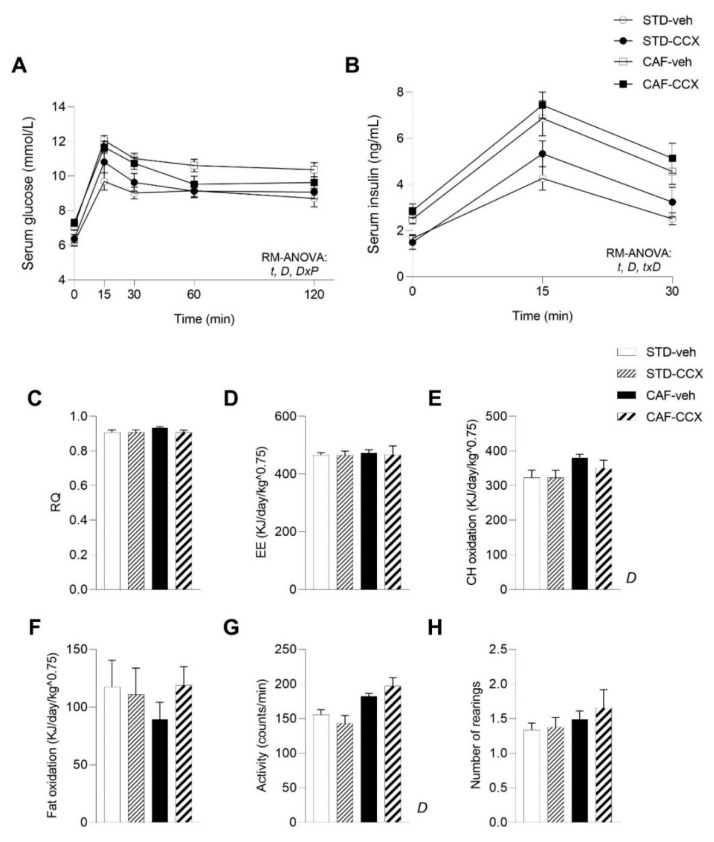
Glucose (**A**) and insulin (**B**) circulating levels after an OGTT (2 g of glucose kg^−1^ of body weight) carried out on postnatal day 80 in male offspring of dams treated with CCX or vehicle during lactation. Respiratory quotient (RQ) (**C**), energy expenditure (EE) (**D**), carbohydrate (CH) oxidation (**E**), fat oxidation (**F**), locomotor activity (**G**) and number of rearings (**H**) obtained by indirect calorimetric measurements at day 83–84 of age of male offspring are also shown. Data are the mean ± SEM (*n* = 10–13). For the plasma glucose and insulin concentrations: *t*: effect of time; *D*: effect of the type of diet; *DxP*: interaction between diet and metabolic programming effect of CCX; *txD*: interaction between time and diet (*p* < 0.05, RM-ANOVA). *D*: effect of the type of diet for the indirect calorimetry-related parameters (two-way ANOVA, *p* < 0.05). STD, chow diet; CCX, cocoanox extract; CAF, cafeteria diet.

**Figure 4 nutrients-14-05134-f004:**
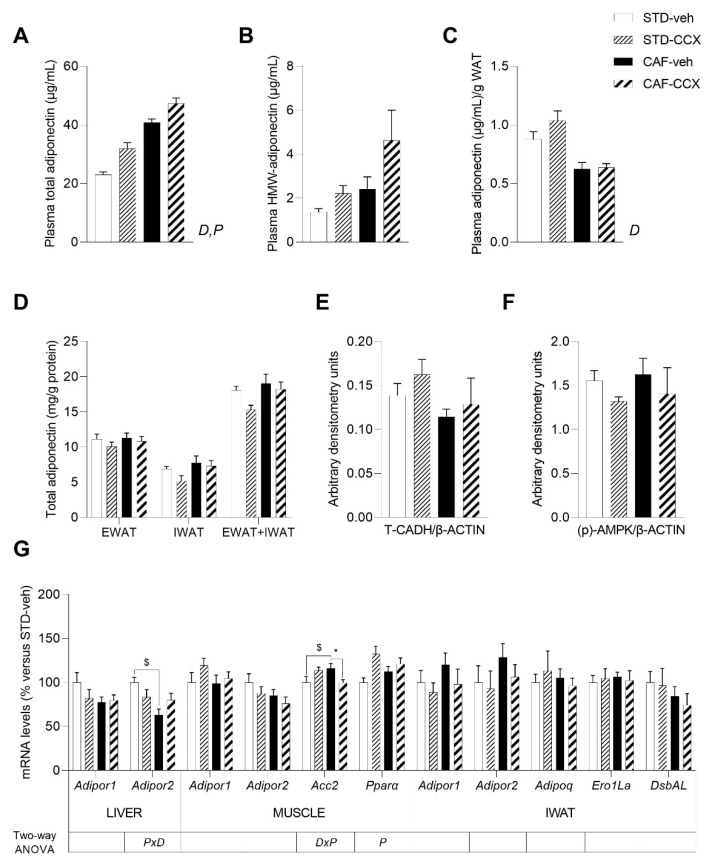
Circulating levels of total adiponectin (**A**), HMW-adiponectin plasma levels (**B**) and adiponectin/WAT weight ratio (**C**) in male offspring of dams treated with CCX or vehicle during lactation. Total adiponectin content in EWAT and IWAT depots (**D**), the protein levels of T-cadherin (T-CADH) (**E**) in gastrocnemius muscle and phosphorylated (p-) AMPK (**F**) in the soleus muscle, as well as the expression levels of adiponectin and lipid metabolism-related genes in liver, soleus muscle and IWAT (**G**) are also shown. Total adiponectin/g WAT was computed as the quotient between the plasma total adiponectin levels and the sum of the EWAT, MWAT, RWAT and IWAT depot weights. Data are the mean ± SEM (*n* = 10–13) for the plasma total adiponectin, adiponectin/WAT weight ratio and adiponectin content in white adipose tissue depots and the mean ± SEM (*n* = 8) for the other analyses. *D*, effect of the type of diet; *P*: metabolic programming effect of CCX; *DxP*: the interaction between diet and metabolic programming effect of CCX (two-way ANOVA, *p* < 0.05). $ The effect of diet within vehicle groups; * metabolic programming effect of CCX within STD groups (Student’s *t* test, *p* < 0.05). STD, chow diet; veh, vehicle; CCX, cocoanox extract; CAF, cafeteria diet; HMW, high molecular weight; WAT, white adipose tissue; EWAT, epididymal white adipose tissue; IWAT, inguinal white adipose tissue; T-CADH, T-cadherin; β-ACTIN, beta-actin; (p)-AMPK, phosphorylated AMP-activated protein kinase; *Adipor1*, adiponectin receptor 1; *Adipor2*, adiponectin receptor 2; *Acc2*, acetyl CoA carboxylase 2; *Pparα*, peroxisome proliferator-activated receptor alpha; *Adipoq*, adiponectin; *Ero1-Lα*, endoplasmic reticulum oxidoreductin 1-like protein alpha; *DsbAL*, disulfide-bond-A oxidoreductase-like protein.

**Figure 5 nutrients-14-05134-f005:**
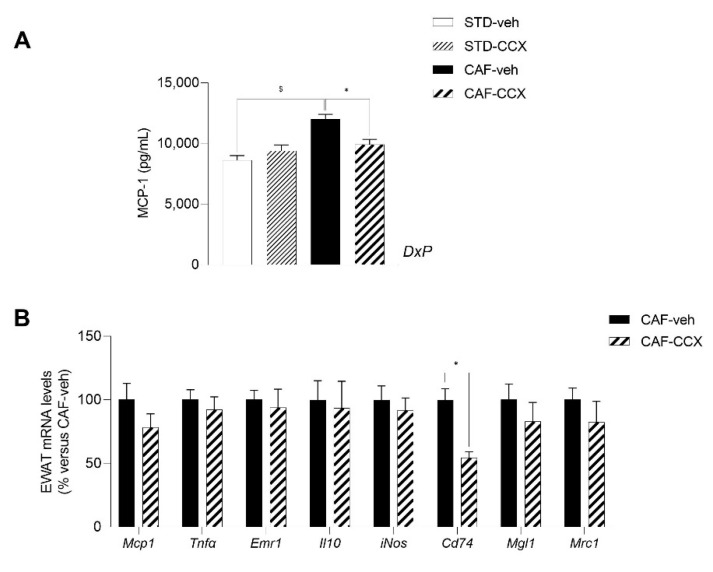
Circulating levels of MCP-1 (**A**) in male offspring of dams treated with CCX or vehicle during lactation. The expression levels of inflammatory response-related genes in EWAT (**B**) are also depicted for the CAF-fed progeny. Data are the mean ± SEM (*n* = 11–13). *DxP*: the interaction between diet and metabolic programming effect of CCX (two-way ANOVA, *p* < 0.05). ^$^ The effect of diet within vehicle groups (Student’s *t* test, *p* < 0.05); * metabolic programming effect of CCX within CAF groups (Student’s *t* test, *p* < 0.05). STD, chow diet; veh, vehicle; CCX, cocoanox extract; CAF, cafeteria diet; *MCP-1*, monocyte chemoattractant protein-1; EWAT, epididymal white adipose tissue; *Tnfα*, tumor necrosis factor; *Emr1*, EGF-like module containing, mucin-like, hormone receptor-like 1; *Il10*, interleukin 10; *iNos*, nitric oxide synthase 2, inducible; *Cd74*, major histocompatibility complex, class II invariant chain; *Mgl1*, C-type lectin domain family 10, member A; *Mrc1*, mannose receptor, C type 1.

**Table 1 nutrients-14-05134-t001:** Biometric, plasma parameters and energy intake of dams fed with chow and treated with vehicle or CCX during lactation.

	VEH	CCX	Student’s *t*-Test (*p*)
**Biometric parameters**			
Initial body weight (g)	278 ± 3	278 ± 6	0.926
Final body weight (g)	286 ± 3	283 ± 5	0.561
Body weight gain (g)	8.00 ± 3.78	5.11 ± 5.28	0.662
RWAT (g)	3.15 ± 0.41	3.18 ± 0.39	0.954
MWAT (g)	1.87 ± 0.14	1.67 ± 0.17	0.388
OWAT (g)	4.93 ± 0.64	4.77 ± 0.39	0.830
Adiposity index (%)	3.81 ± 0.45	3.83 ± 0.39	0.977
Mammary gland (g)	11.0 ± 0.3	11.7 ± 0.3	0.096
Liver (g)	10.4 ± 0.1	10.0 ± 0.2	0.129
Lean mass (g)	249 ± 4	243 ± 6	0.452
Lean mass (%)	86.3 ± 0.7	85.9 ± 0.7	0.725
Fat mass (g)	23.1 ± 1.6	23.0 ± 2.9	0.987
Fat mass (%)	8.01 ± 0.59	8.20 ± 1.07	0.885
**Plasma parameters**			
Glucose (mmol/L)	5.93 ± 0.15	5.57 ± 0.24	0.228
Insulin (mU/L)	43.9 ± 6.6	57.9 ± 19.7	0.516
FFAs (mmol/L)	0.218 ± 0.025	0.348 ± 0.043 *	**0.020**
HOMA-IR	11.7 ± 2.0	15.5 ± 5.5	0.530
R-QUICKI	0.338 ± 0.007	0.339 ± 0.022	0.973
Total adiponectin (µg/mL)	17.6 ± 0.7	23.9 ± 1.9 *	**0.010**
HMW adiponectin (μg/mL)	2.99 ± 0.55	3.93 ± 1.03	0.401
Total adiponectin/g WAT	1.72 ± 0.17	2.31 ± 0.20 *	**0.041**
Triglycerides (mmol/L)	0.375 ± 0.024	0.353 ± 0.029	0.547
Phospholipids (mmol/L)	2.11 ± 0.07	2.03 ± 0.08	0.489
Total cholesterol (mmol/L)	2.21 ± 0.11	1.96 ± 0.07	0.065
**Energy intake (kcal/day)**	155 ± 2	147 ± 5	0.148

Dams were orally supplemented with CCX (200 mg.kg^−1^.day^−1^) (CCX group) or vehicle (VEH group) during lactation (21 days). Data are the mean ± SEM (*n* = 9). * *p* < 0.05 vs. VEH group (Student’s *t*-test). CCX, cocoanox extract; RWAT, retroperitoneal white adipose tissue; MWAT, mesenteric white adipose tissue; OWAT, ovarian white adipose tissue; FFAs, non-esterified free fatty acids; HOMA-IR, Homeostatic model assessment of insulin resistance; R-QUICKI, revised quantitative insulin sensitivity check index; HMW, high molecular weight; WAT, white adipose tissue. Total adiponectin/g WAT was computed as the quotient between the circulating levels of total adiponectin and the sum of the OWAT, MWAT and RWAT depot weights.

**Table 2 nutrients-14-05134-t002:** Biometric and plasma parameters, and energy intake in 90-day-old male offspring of rats treated with CCX or vehicle during lactation.

	STD-veh	STD-CCX	CAF-veh	CAF-CCX	Two-Way ANOVA
**Biometric parameters**					
Initial body weight (g)	39.7 ± 1.9	41.3 ± 1.3	41.0 ± 1.4	40.8 ± 1.1	*-*
Final body weight (g)	373 ± 9	371 ± 4	446 ± 12	433 ± 14	*D*
Body weight gain (g)	333 ± 8	330 ± 4	405 ± 12	392 ± 14	*D*
Liver (g)	12.0 ± 0.4	10.9 ± 0.3	14.3 ± 0.5	13.2 ± 0.4	*D*, *P*
RWAT (g)	6.47 ± 0.45	7.87 ± 0.61	18.6 ± 1.7	19.0 ± 1.4	*D*
MWAT (g)	4.28 ± 0.27	4.75 ± 0.33	8.46 ± 0.57	9.21 ± 0.80	*D*
EWAT (g)	5.67 ± 0.48	6.86 ± 0.57	14.9 ± 1.3	15.8 ± 1.5	*D*
IWAT (g)	11.9 ± 1.0	12.4 ± 1.0	29.2 ± 3.0	30.0 ± 2.4	*D*
Adiposity index (%)	7.75 ± 0.54	8.69 ± 0.61	16.1 ± 1.0	17.2 ± 0.9	*D*
Soleus muscle (g)	0.147 ± 0.006	0.135 ± 0.005	0.142 ± 0.004	0.134 ± 0.002	*P*
Gastrocnemius muscle (g)	1.74 ± 0.05	1.76 ± 0.03	1.86 ± 0.02	1.79 ± 0.03	*D*
Lean mass (%)	86.2 ± 0.8	85.0 ± 1.0	74.8 ± 1.4	73.2 ± 1.2	*D*
Fat mass (%)	8.64 ± 0.76	9.86 ± 1.04	20.5 ± 1.4	22.4 ± 1.3	*D*
Lean/fat ratio	10.8 ± 0.9	8.62 ± 0.81	3.94 ± 0.35	3.46 ± 0.29	*D, P*
**Plasma parameters**					
Glucose (mmol/L)	6.00 ± 0.34	6.09 ± 0.25	7.41 ± 0.34	7.72 ± 0.13	*D*
Insulin (mU/L)	77.2 ± 8.4	94.9 ± 10.3	127 ± 10	141 ± 8	*D*
HOMA-IR	20.4 ± 2.5	26.3 ± 3.8	43.0 ± 5.2	48.4 ± 3.0	*D*
R-QUICKI	0.306 ± 0.008	0.291 ± 0.004	0.269 ± 0.004	0.270 ± 0.003	*D*
Total cholesterol (mmol/L)	1.65 ± 0.11	1.68 ± 0.07	2.20 ± 0.09	2.09 ± 0.08	*D*
Triglycerides (mmol/L)	1.75 ± 0.32	1.15 ± 0.22	3.67 ± 0.39	3.17 ± 0.25	*D*
Free fatty acids (mmol/L)	0.279 ± 0.035	0.272 ± 0.030	0.336 ± 0.022	0.280 ± 0.022	*-*
Phospholipids (mmol/L)	1.77 ± 0.09	1.72 ± 0.08	2.38 ± 0.12	2.51 ± 0.11	*D*
Leptin (ng/mL)	9.33 ± 0.63	9.89 ± 0.77	39.6 ± 3.6	45.4 ± 3.1	*D*
Adiponectin/leptin ratio	2.65 ± 0.22	3.40 ± 0.31 *	1.18 ± 0.16 ^$^	1.09 ± 0.07	*D, P, DxP*
**Energy intake (kcal/day)**	52.8 ± 1.5	55.3 ± 1.3	112 ± 3	113 ± 3	*D*

Ninety day-old male offspring of dams that received an oral dose of CCX (200 mg.kg^−1^.day^−1^) or a vehicle during lactation (21 days). The offspring was fed with either a STD or a CAF from postnatal day 21 until the age of 90 days, which led to four experimental groups: STD-veh, STD-CCX, CAF-veh and CAF-CCX. Data are the mean + SEM (*n* = 11–13). *D*: effect of the type of diet; *P*: metabolic programming effect of CCX; *DxP*: the interaction between diet and metabolic programming effect of CCX (two-way ANOVA, *p* < 0.05). ^$^ The effect of diet within vehicle groups (Student’s *t* test, *p* < 0.05); * metabolic programming effect of CCX within STD groups (Student’s *t* test, *p* < 0.05). STD, chow diet; CCX, cocoanox extract; CAF, cafeteria diet; RWAT, retroperitoneal white adipose tissue; MWAT, mesenteric white adipose tissue; EWAT, epididymal white adipose tissue; IWAT, inguinal white adipose tissue; HOMA-IR, Homeostatic model assessment of insulin resistance; R-QUICKI, revised quantitative insulin sensitivity check index.

## Data Availability

Not applicable.

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
