# Peer review of "Maternal Supplementation with a Cocoa Extract during Lactation Deeply Modulates Dams’ Metabolism, Increases Adiponectin Circulating Levels and Improves the Inflammatory Profile in Obese Rat Offspring"

_nutrients, 2022, doi:10.3390/nu14235134_

Round 1

Reviewer 1 Report

The authors present a generally well-written article on an interesting scientific question. Over the last decades it has become clear that early life nutrition influences health throughout life. Due to numerous confounding factors it is very difficult to study the impact of single substances in early life nutrition on long-term health in humans. Therefore, animal studies are an alternative approach.

The animal design used is clearly described and appropriate to answer the scientific questions. However, I have some suggestions:

1) Most intriguingly, cocoa extract has a significant impact on maternal and offspring adiponectin levels, mammary gland lipids and circulating MCP-1 levels. Of minor importance, the authors also performed numerous analyses of gene expressions. As a consequence, both the results part and the dicussion part are very long. However, differences found on the gene expression level do not necessarily correspond to the protein levels of the respective genes. In addition, in some cases the tissue used for RNA analyses does not correspond to the protein data discussed (e.g. circulating MCP-1, CD74 in white adipose tissue) which is a little confusing. Therefore, I would suggest to focus more clearly on the protein/lipid results and present RNA data as an overview. To clearly answer the question of inflammatory cell infiltration in white adipose tissue more analyses are needed.

2) Lipids: The altered composition of mammary gland composition is very interesting. Is it possible to analyze the n3/n6 ratio of the lipids?

3) Although no graphical abtract is needed, I would appreciate a figure which summarizes the design and the main results like a graphical abtract would do.

Reviewer 2 Report

Thank you for the privilege of reviewing the research and the manuscript. In this work, the authors presented the maternal cocoa intake during lactation which may be associated with the health of dams and their male offspring fed with different food.

1) For CCX-treated dams: the circulating levels of adiponectin and FFA and energy expenditure were increased, as well as a higher expression of lipogenic and adiponectin-related genes in the mammary gland.

2) For offspring: increased plasma total adiponectin levels and decreased liver weight and lean/fat ratio were observed in the offspring of lactating dams. The authors also found that the CAF-CCX progeny showed an improvement in the inflammatory profile. These results suggested the beneficial effects of cocoa consumption during lactation for dams and progeny.

Overall, the method and results were well described, but there are some minor points.

Minor points

1. Please revise your introduction and make it more concise. I suggest shortening the description of obesity in lines 36-51. Also, in lines 59-87, it would be better for understanding to introduce cocoa first and then polyphenol......

2. In lines 92-99, you mentioned that the cocoa catechins administered to pregnant mice cause deleterious effects in the offspring. Is there any evidence supporting your hypothesis that the maternal intake of a cocoa extract during lactation brings positive effects to dams and their offspring?

3. Please explain why you choose 200 mg.kg-1.day-1 as maternal cocoa extract consumption. It would be better to explain this with some references or some experimental data.

4. Please add p values in Table 1.

5. Line 456-473, it would bebetter to show key results.

6. In lines 507-515, the authors concluded the effects on dams brought by supplementation with cocoa extract during lactation. However, the authors didn’t further summarize the results for offspring.

7. Line 525-527, the authors stated that the inflammatory molecules act as negative regulators of adiponectin secretion. But in lines 704-706, you said the increased circulating levels of adiponectin responsible for the MCP-1 plasma levels decrease.   Seems contradictory.

8. In this work, it is important to provide evidence to illustrate the transmission mechanism of how the dam’s metabolic programming could affect the offspring's health in the discussion part. 

9. I suggest shortening the discussion section. It should be revised to be more concise to highlight the key points.

10. Please modify the table format according to journal’s requirements and previous articles published in Nutrients. The titles and legends are misaligned in Table1 and Table 2.

11. Line 282-286, 301-307, and 355-362, the abbreviations should be presented according to the order they appeared in the tables and figures.
